# Antimicrobial Performance of Innovative Functionalized Surfaces Based on Enamel Coatings: The Effect of Silver-Based Additives on the Antibacterial and Antifungal Activity

**DOI:** 10.3390/ijms24032364

**Published:** 2023-01-25

**Authors:** Hamza Ur Rehman, Francesca Russo, Massimo Calovi, Orietta Massidda, Stefano Rossi

**Affiliations:** 1Department of Cellular, Computational and Integrative Biology, University of Trento, 38123 Trento, Italy; 2Department of Industrial Engineering, University of Trento, 38123 Trento, Italy

**Keywords:** vitreous enamel, antimicrobial coatings, silver, fungicidal activity

## Abstract

Frequently touched surfaces (FTS) that are contaminated with pathogens are one of the main sources of nosocomial infections, which commonly include hospital-acquired and healthcare-associated infections (HAIs). HAIs are considered the most common adverse event that has a significant burden on the public’s health worldwide currently. The persistence of pathogens on contaminated surfaces and the transmission of multi-drug resistant (MDR) pathogens by way of healthcare surfaces, which are frequently touched by healthcare workers, visitors, and patients increase the risk of acquiring infectious agents in hospital environments. Moreover, not only in hospitals but also in high-traffic public places, FTS play a major role in the spreading of pathogens. Consequently, attention has been devoted to developing novel and alternative methods to tackle this problem. This study planned to produce and characterize innovative functionalized enameled coated surfaces supplemented with 1% AgNO_3_ and 2% AgNO_3_. Thus, the antimicrobial properties of the enamels against relevant nosocomial pathogens including the Gram-positive *Staphylococcus aureus* and the Gram-negative *Escherichia coli* and the yeast *Candida albicans* were assessed using the ISO:22196:2011 norm.

## 1. Introduction

With the increasing number of infectious diseases, their management has become a growing challenge. According to the Centers for Diseases Control and Prevention (CDC) and the World Health Organization (WHO), infectious diseases are a primary global concern [1].

Contaminated surfaces within healthcare and public environments play a significant role as a potential driver in the transmission of infectious diseases [2,3,4]. Since January 2020, with the emergence and spread of the SARS-CoV-2 pandemic, hygiene and precautionary measures such as sanitizing hands and disinfection of surfaces have been implemented worldwide at all levels [5,6].

Pathogenic microorganisms can survive for a prolonged time on frequently touched surfaces (FTS) and are the leading cause of outbreaks of hospital-acquired infections and healthcare-associated infections (commonly named HAIs) [7]. HAIs are the most frequent affliction [8] that affect between 5% and 10% of patients admitted to hospitals in developed countries, of which 15–40% end up in intensive care units (ICU). The risk of contracting HAIs is much higher in developing countries [9]. According to the European Center for Disease Prevention and Control (ECDC), in Europe HAIs are acquired by more than 4 million people annually compared to 1.7 million in the US [10]. This is because pathogens typically shed by healthcare workers and patients’ hands contaminate the surrounding surfaces for days, increasing the risk of transmission and acquisition by healthy individuals and other patients [11]. Therefore, along with the routine practice of hand hygiene with sanitizers and alcohol-based hand rub, environmental hygiene and disinfection is another crucial aspect in controlling these infections [12,13].

Ethanol is the most common alcohol ingredient used routinely as a disinfectant and appears to have both effective antibacterial and antiviral potential. However, ethanol does not seem very effective against acquired antibiotic-resistant bacteria and, most importantly, must be used repeatedly in medical settings [14]. In addition, or as an alternative to the use of disinfectants, novel strategies are required [11,15], as the microorganisms causing HAIs also show resistance due to biofilm formation, which renders them phenotypically less susceptible to disinfectants [1] and to a large number of antimicrobials used in therapy [16]. Furthermore, unhygienic environmental surfaces elevate the resistance of pathogens to disinfectants. In parallel to FTS at a nosocomial level that is responsible for HAIs, the role of surfaces in the transmission of pathogenic microorganisms in public settings, including public transport [17], public toilets [18], and even kitchens and washrooms in a home [19] should not be neglected where doorknobs, lift buttons and light switches serve as a reservoir of a large number of pathogens [20]. The persistency of pathogens on FTS for months [21] makes it essential to adopt and carry out routine cleaning of these surfaces through non-alcohol-based disinfectants [22], that persist for more extended periods, or through the development of functionalized surfaces/materials with antimicrobial activity to counteract the spread of pathogens.

Antimicrobial coatings represent an effective and advancing strategy against microbial growth and probably the development of multidrug-resistant bacteria to reduce HAIs [20,23,24]. Antimicrobial materials create self-disinfecting surfaces by applying coatings with antimicrobial activity. Antimicrobial materials are divided into anti-biofouling surfaces (also known as non-adhesive surfaces) that prevent microbial adhesion and actively antimicrobial surfaces, which are further grouped into biocide-releasing surfaces that contain active eluting agents (such as ions silver, copper, zinc, or antibiotics, chloride, iodine), contact killing surfaces that become active upon contact of microbes (such as quaternary ammonium polymers or peptides), and light-activated antimicrobial surfaces that require photosensitizers such as TiO_2_ photocatalyst thin films [23,25]. In addition to chemical additions, the surface topographical properties (e.g., the durability of material and prevention of microbial retention over time) should always be considered, which can considerably affect hygienic properties [26].

The property of high durability is particularly appreciated in vitreous enamel [27]. Vitreous enamel is an inorganic coating that possesses excellent physical and mechanical properties, such as resistance to light, scratches, chemicals, intense temperature, abrasion, corrosion, optimal ease of cleaning, and hygiene. These properties make it a suitable candidate for use in public healthcare and high-traffic environments where frequent cleaning and disinfection are compulsory [28]. Moreover, despite being based on a relatively ancient production technique, vitreous enamel coatings meet very modern criteria of environmental friendliness and recyclability. At the end of life, enameled metallic components can be treated as metal scraps, without the need to separate the coating from the metal substrate, as enamel compounds become smelt waste that separates from the metal and does not alter the properties of the smelt itself [29]. In addition to that, many recent technological advances have allowed the improvement of closed-loop waste management in the enameling process. In this context, reducing harmful chemicals and introducing new deposition methods have allowed the full recycling of effluents and the recycling of wet and dry formulations with a consequent reduction in over spraying [30,31]. Thus, the production of enamel coatings is perfectly compatible with the efforts to promote the circular economy and the closed-loop management plan.

Silver-based antimicrobial enamel coatings have been recently exploited [20]: these materials could represent a solution to prevent the retention, spread, and transmission of pathogenic microorganisms at the nosocomial level and in public high-traffic environments.

Ag^+^ ions show the highest antimicrobial potential that has been demonstrated in a wide range of applications against bacteria, fungi, protozoa, and certain viruses [32]. The mechanism of action of Ag^+^ ions against bacteria still needs to be understood entirely. However, several mechanisms have been proposed and accepted (Figure 1).
As bacterial cell walls are made up of negatively charged peptidoglycan (PG), the positively charged Ag^+^ ions will bind to the cell walls, the cell membrane, and the membranal transport proteins, thus blocking the transport of essential substances in and out of the bacterial cells [33,34,35].The Ag^+^ ions may destroy the energy production of the bacterial cell by binding to the cellular proteins and enzymes through sulfhydryl groups (-SH), which leads to the inactivation of proteins and enzymes [33,35,36,37]. Subsequently, the bacterial cell will burst, leading to bacteria destruction [37].Once inside the cell, Ag^+^ ions will bind to bacterial DNA [33] and subsequently block replication and cell division [33,38,39].

Silver has been widely used in the fabrication of antimicrobial surfaces/materials. It is considered one of the best alternatives in antimicrobial coatings due to its substantial antimicrobial effects even at low concentrations [40]. Its addition to polymers allows a persistent control of the environmental pathogenic bacteria [41]. It has been added to plastics [42] and synthetic fabrics [43], with a strong reduction in bacterial contamination. Moreover, some works have reported also the antifungal activity of silver, and silver nanoparticles have been shown to exert activity against dermatophytes [44]. It has been also shown that the conjugation of silver nanoparticles with multipurpose contact lenses and amphotericin B, an antifungal drug, significantly reduces corneal infections caused by *Acanthamoeba castellanii* [45,46]. Furthermore, it has been suggested that Ag^+^ ions may act against fungi either by destroying the integrity of the cell membrane and halting the budding process [44] or through increased production of reactive oxygen species (ROS) after exposure in fungi [47], supporting the notion that silver nanoparticles with different sizes and shapes possess different levels of effectiveness, which in turn vary depending on the fungus and incubation time [48].

This work aimed to test the antimicrobial activities of newly developed innovative antimicrobial functionalized surfaces based on enamel coatings with the addition of silver-based additives (AgNO_3_) that have previously shown efficient antibacterial activity [20]. The antibacterial activity of the composite coatings was tested in a time-course experiment against *Escherichia coli* and *Staphylococcus aureus*, as two main representative causative agents of HAIs within Gram-negative and Gram-positive bacteria, respectively [49]. In addition, the enamel coatings with silver-based additives were also tested against *Candida albicans*, as a model organism for fungal pathogens in hospital settings [50].

## 2. Results

### 2.1. Coatings Characterization

Figure 2 depicts the appearance of the three types of samples. The image illustrates a distinct variation in the surface hue of the different enamels, owing to the different concentrations of silver in the top layers of sample E1 and sample E2, containing 1 wt.% and 2 wt.% of AgNO_3_, respectively. As a result, adding silver alters the visual features of the coatings, changing the upper shade of the materials.

Table 1 shows the roughness evaluation of the samples acquired with the PS1 roughness gauge (5 measurements on five samples—a total of 25 measurements). As demonstrated in Table 1, the surface roughness of each sample has shallow values (comparable among all the samples), and the addition of silver does not show a discernable impact on the topography of the samples that are typical of suitable manufactured enamel coatings.

Cross-sectional analyses were carried out via optical and SEM microscopes to measure the thickness of three different layers (ground coat, white coat, and silver top-coat) and the internal microstructure of the test samples. The measurements in Table 2 relative to the thickness of the different layers are the average value of 5 measurements carried out on five samples (25 measurements in total) per series. The samples were cut and cold-mounted with an epoxy bi-component resin. Then, the samples were ground and mechanically polished from a #180 SiC grinding paper until a 1 µm polishing diamond paste was ready to be observed under optical and SEM microscopes.

Figure 3 represents the optical cross-sectional micrographs of the samples. The discrimination between the three different layers of the test samples is shown in the images. The dark bottom layer is the ground-coat layer and is often placed to ensure proper adhesion between the metal substrate and the enamel coating [51]. The ground-coat layer thickness is approximately 39–51 μm for all samples, as shown in Table 2. The thickness of the white-coat layer displays some inhomogeneity, ranging from 120–151 μm for each sample. This thickness variance is mostly attributed to the inherent unpredictability of the selected deposition technique, which does not guarantee complete reproducibility as it is heavily impacted by operator expertise. Finally, the pink silver-based top-coat represents the outer layer, with an estimated thickness of 30 μm and 24 μm for samples E1 and E2, respectively.

Similarly, Figure 4 represents the SEM micrographs of the three series of samples. The compact layers of the enamels exhibit the typically closed porosity structure which characterizes this type of protective coatings. The concentration of porosity decreases going up from the ground layer to the top-coat. The same bubbles appear larger (about 50 µm) in the ground and then decrease in size in the overlaying two layers.

### 2.2. In Vitro Analysis of Coatings Antimicrobial Effectiveness

#### 2.2.1. Antibacterial Efficacy

The treated enameled surfaces reduced the number of both *E. coli* and *S. aureus* strains compared to their respective untreated controls after 2, 8, and 24 h of exposure.

Figure 5 shows the reduction of the CFU/mL after the exposure of the silver-treated contaminated surfaces with *E. coli*. While the reference sample EE0 showed a relatively low decrease in CFU/mL over time, the presence of silver in the two samples, EE1, and EE2, caused a clear drop in the CFU/mL during the time course of the experiment. This result was more evident at higher silver concentrations, with no *E. coli* cells recovered in the EE2 sample already after 8 h of exposure.

To better highlight the inhibitory activity of silver, Figure 6 shows the percentage of recovered bacteria over time, normalized with respect to each untreated reference sample taken at the same time. The treated samples EE1 reduced the growth of *E. coli* to 30% and 98%, after an exposure time of 2 and 8 h, respectively, while the treated samples EE2 completely reduced the number of *E. coli* cells after 8 h. Both silver concentrations, however, resulted in complete inhibition of *E. coli* after 24 h, in agreement with our previous report [20]. The antibacterial activity (R) of the Ag-coated enameled samples against *E. coli* at 24 h, calculated based on the ISO:22196 formula, was of 3.55 for both EE1 and EE2.

Similarly, Figure 7 shows the CFU/mL recovered after the exposure of the contaminated surfaces with *S. aureus*. In this case, the antibacterial activity of silver was not particularly evident before 8 h of exposure for any of the silver-treated samples. However, sample SE2, containing a higher amount of Ag, showed a more rapid inhibitory behavior than sample SE1.

Figure 8, which illustrates the percentage of recovered bacteria over time, normalized with respect to each untreated reference sample taken at the same time, clearly points out this. Accordingly, the number of *S. aureus* cells was reduced in sample SE1 by 6% and 20% after 2 and 8 h of exposure, respectively. On the other hand, sample SE2 showed a 68% reduction compared to the untreated samples SE0 already after 8 h of exposure. Both silver concentrations, however, resulted in complete inhibition of *S. aureus* after 24 h, in agreement with our previous report [20]. The antibacterial activity (R) of the Ag-coated enameled samples against *S. aureus* at 24 h, calculated based on the ISO:22196 formula, was of 3.14 for both SE1 and SE2.

#### 2.2.2. Antifungal Activity

A protocol similar to that used to evaluate the antibacterial activity of silver enamel against *E. coli* and *S. aureus* was used to analyze the antifungal potential of the treated enameled surfaces against *C. albicans*. *C. albicans* is an opportunistic dimorphic fungus that can grow as both yeast and filamentous cells forms; it has a commensal relationship with humans, but it is also responsible for about 70% of oral infections in humans and about 75% of genital infections in women [49,52]. Figure 9 shows the typical morphology of *C. albicans* cells, observed with the optical microscope after 24 h of contact with the surface of the reference sample E0.

Figure 10 reveals the reduction of the number of CFU/mL recovered during the exposure of the samples to *C. albicans*. In this case, due to the slower growth of *C. albicans* with respect to bacteria, the exposure was monitored every 24 h up to 72 h, to better observe the antifungal effect of silver, which became more evident over time.

As expected, the CFU/mL of the reference sample CE0 increased over time, as *C. albicans* requires longer incubation times for growth with respect to bacteria. Conversely, silver causes a sharp decrease in CFU/mL after 48 h in the case of sample CE1, and already after 24 h in the case of sample CE2. Again, the amount of silver seemed to play a role in the antimicrobial kinetics, anticipating a more efficient antifungal activity at higher Ag concentrations.

Figure 11 shows the percentage (%) of recovered fungal cells from three surfaces of each sample at each time point, normalized with respect to each untreated reference sample taken at the same time. Both CE1 and CE2 showed a reduction in the recovery of *C. albicans*. After 24 h of contact with CE1 and CE2, *C. albicans* showed a reduction of 80% and 98%, respectively. Longer exposure (48 h and 72 h) with CE1 and CE2 led to a larger inhibitory effect on *C. albicans* with a 99.9% reduction compared to the untreated surface CE0. Consistent with these results, the R activity for CE1 and CE2, based on the ISO:22196 formula, was of 1.53 and 1.88, respectively, at 24 h, of 2.19 and 3.03, respectively, at 48 h to reach 3.68 and 3.71, respectively, at 72 h.

## 3. Discussion

The deposition of the silver layer in the most superficial layers of the coating proved to be able to guarantee an effective antimicrobial activity for longer exposure times [20]. The EDXS analyses carried out on these outer layers in previous work [20] demonstrated that sample E1 and sample E2 contain 0.5 wt.% and 1.5 wt.% of silver, respectively. Furthermore, from the SEM images, which highlight the internal morphology of the coatings, the presence of silver in the top-coat does not alter the structure of the enameled layer, nor does it preclude its correct deposition.

The results of the current work confirm that the functionalized surfaces, such as enameled coated surfaces supplemented with 1% wt. AgNO_3_ (E1) and 2% wt. AgNO_3_ (E2), possesses a remarkable antibacterial activity against the Gram-negative *E. coli* and the Gram-positive *S. aureus* and demonstrated that this activity occurs rapidly over time, dependent on AgNO_3_ concentrations. Indeed, both E1 and E2 showed a consistent reduction in the number of *E. coli* after 2 h of exposure to the surfaces. Duration of 8 h and 24 h of contact with the treated surfaces led to complete inhibition of *E. coli*. On the other hand, *S. aureus* showed somewhat more resistance to both E1 and E2 after 2 and 8 h of contact. However, the complete absence of recovery of *S. aureus*, observed after 24 h of contact with the treated surfaces, is consistent with the antibacterial activity observed in a recent work published by Russo et al. [20]. As sample E2 possesses a higher concentration of Ag, it exhibits a more rapid antibacterial effect against both the Gram-negative bacteria and Gram-positive tested in this work and is consistent with the norm for these studies [53,54,55].

The antibacterial activity of both AgNO_3_ concentrations was very consistent in all the experiments performed with *E. coli*, whereas the experiments performed with *S. aureus* showed more fluctuation. It has been shown that Gram-negative bacteria are more vulnerable to silver ions than Gram-positive bacteria [55], which is in agreement with the results in the current work. This may be mainly because the cells wall of Gram-positives and Gram-negatives differ significantly. The layers of PG in the cell wall of Gram-positives are thicker than those surrounding the Gram-negatives [56], likely enforcing a mechanism of protection against silver ions’ entry into the cytoplasm [36]. However, no bacterial growth or multiplication was observed during continuous cultivation by inoculating the silver-treated cells into a fresh liquid LB media according to a work published by Feng et al. [36].

According to the concept of fungistatic and fungicidal action [57], both silver-based enameled coatings were shown to have a fungistatic effect after 24 h and a likely fungicidal effect after 48 and 72 h against *C. albicans*.

A few studies have demonstrated the antifungal properties of AgNPs against *C. albicans* [47,58,59,60,61], but no data on the antifungal properties of silver-based enameled coatings are available. The finding of the present work is consistent with the finding of previous studies that the effect of silver against fungi is dependent on its concentration [62] and incubation time [63], which explains why E1 is less active after 24 h and 48 h than E2. The current understanding of the mode of action of silver against fungi is poor but ultrastructure analysis of *C. albicans* while exposed to AgNPs revealed particles cluster outside the fungal cells, slowly releasing the silver ions and driving cell death via the reduction process by the contact of cell components with ionic silver [58]. As previously mentioned, a 99.9% reduction in the viability of *C. albicans* was observed after 48 h of contact time by both E1 and E2. The results were similar after 72 h, indicating an antifungal effect that fails to reach 100% efficacy in such a short time.

## 4. Material and Methods

### 4.1. Samples Deposition and Morphological Characterization

The samples were produced according to the previous work [20]. DC04EK cold-rolled steel panels (40 mm × 40 mm × 1.5 mm) were used as substrates for the coatings deposition. The steel panels were degreased in a hot detergent solution at 40 °C for 10 min, to remove possible contaminants and activate the surface for the deposition step. The frit and the samples were developed and deposited at Emaylum Italia s.r.l (Chignolo d’Isola, BG, Italy). Silver nitrate (AgNO_3_ > 99.0%, Sigma-Aldrich, St. Louis, MO, USA) was employed as the source of silver. Three series of samples were produced by the wet spraying method: the reference sample, free of silver, and the sample with the addition of 1 wt.% AgNO_3_, and the sample with the addition of 2 wt.% AgNO_3_. The reference sample was deposited by a 2A/2F (2 applications—2 firings) cycle: a “ground coat”, was fired at 850 °C for 3–4 min, and subsequently, a “white top-coat”, fired at 800 °C for 2–3 min, was deposited on it. The firing steps were carried out in a furnace in an air atmosphere. In order to limit the interpenetration of the two layers during the second firing step, the top-coat was fired at a lower temperature. The two samples containing silver were deposited by a 3A/3F application cycle: the three layers—namely, ground coat, white top-coat, and silver-based coat—were deposited and fired in separate steps. The silver-based layer was fired at 800 °C for 2–3 min. The formulation of the single layers can be found in the previous work [20]. Table 3 shows the nomenclature of the three series of samples.

As a first step, the topological parameters of the enamel coatings, such as surface roughness and thickness, were characterized. The roughness of the surfaces was characterized using the MAHR Marsurf PS1 (MAHR GmbH, Göttingen, Germany) roughness gauge as per ISO 4287:1997 standard [64]. The Ra and Rz values exhibited in the results represent the averages of 25 measurements performed on five different surfaces from each sample. The thickness of the samples, including the thickness of different layers (ground-coat, white-coat, and silver-coat layers), was analyzed with the stereotypical optical microscope (NIKON SMZ25, Nikon Instruments, Amstelveen, the Netherlands). A total of 50 measurements were taken on five surfaces from each sample. The internal microstructure of the antimicrobial coatings was exploited using the stereotypical optical microscope (NIKON SMZ25, Nikon Instruments, Amstelveen, the Netherlands) and scanning electron microscope (SEM JEOL IT300, JEOL, Akishima, Tokyo, Japan) in low-vacuum mode at 20 kV.

### 4.2. Microbial Strains and Culture Media

The microorganisms tested in this work were the Gram-positive *S. aureus* ATCC 6538 strain, the Gram-negative *E. coli* ATCC 8739 strain, and the dimorphic fungus *C. albicans*. Nutrient agar (NA), Nutrient broth (NB), Sabouraud dextrose agar (SDA), and Plate count agar (PCA) were purchased from Microbial S.n.c (Uta, Cagliari, Italy). Tryptone soya agar (TSA) and Tryptone soya broth (TSB) were purchased from Oxoid (Oxoid Ltd. Basingstoke, Hants, UK). These media were used for bacterial and fungal stock culture preparation, pre-culture, inoculum preparation and to calculate the viable counts in all antimicrobial assays. Phosphate-buffered saline (PBS 1×), obtained from Gibco (Life Sciences, Waltham, MA, USA), was used for bacterial and fungal cell recovery from the test specimens and for 10-fold serial dilutions.

Bacterial stock cultures were kept in TSB at −80 °C with 15% glycerol for Gram-positives and with 30% glycerol for Gram-negatives. Fungal stock cultures were stored in sterile distilled water with 20% glycerol. All media were prepared as per the manufacturer’s instructions.

### 4.3. Evaluation of Antimicrobial Activity

The antimicrobial efficacy of silver-based enameled coated surfaces at different interval points was evaluated using the ISO 22196:2011 standard [65], with some modifications when necessary. The steps carried out during this study are summarized in Figure 12.
Sterilization of test specimens and cover films: The test samples (reference, 1% AgNO_3_ coated, and 2% AgNO_3_ coated surface) and cover films were sterilized following the protocol described by Calovi et al. [66]. The polypropylene (PP) films (3.6 × 3.6 cm) were used to cover the suspension of test microbes dispersed on coated surfaces (4.0 × 4.0 cm) to prevent evaporation of microbial suspension. PP films and test specimens were kept in 70% ethanol for 10 min and then exposed to ultraviolet (UV) radiation for 1 h per sample side inside the laminar flow hood for sterilization.Preparation of the test *inoculum*: The microbial strains were sub-cultured on appropriate agar plates (NA for *E. coli*, TSA for *S. aureus*, and SDA for *C. albicans*) at 37 °C for 24–48 h, according to the microbial strain, to obtain fresh cultures. Isolated colonies of the test microorganisms from the fresh cultures were resuspended in 10 mL of 1:500 NB (*E. coli* and *S. aureus*) or TSB (*C. albicans*) using a sterile loop. Cell number was adjusted to ~1.5 × 10^8^ cells/mL, corresponding to 0.5 McFarland standard, and the optical density (OD) at 600 nm (OD600), ranging between 0.175 and 0.180 for *E. coli*, 0.210 for *S. aureus* and 0.250 for *C. albicans*, was determined with a UV-Vis spectrophotometer (Biochrom, Cambridge, USA). The suspensions were then diluted in PBS 1× to obtain the respective inoculum with a final concentration ranging from 3.9 × 10^5^ cells/mL to 1.6 × 10^6^ cells/mL (target concentration of 9.4 × 10^5^ cells/mL). Microbial counts were determined by measuring the colony-forming units per mL (CFU/mL), following 10-fold dilution in PBS 1× and inoculated onto agar plates using an L-shaped loop and incubated for 24–48 h at 37 °C.Inoculation of the test specimens: The sterilized enameled surfaces were placed in the middle of the petri dish, with the coated surface facing up. The test inoculum was pipetted onto each surface of the test samples. The contaminated surfaces were covered with a PP film, gently pressing down to spread the test inoculum on the coated surfaces but not to leak beyond the edges of the PP films. Then, the contaminated surfaces were incubated for different time intervals at 35 ± 1 °C, keeping the relative humidity above 90%. Reference samples used as controls to represent the test inoculum were processed instantly after microbial suspension at the T0.Recovery of microbial cells from test specimens: Microorganisms were recovered from each sample after different inoculation times by adding 10 mL of PBS 1× to the Petri dishes containing the contaminated samples. Upon adding 10 mL of PBS 1×, the Petri dishes containing the contaminated samples were placed on Major Science Orbital Shaker (Saratoga, CA, USA) for 5 min at 125 revolutions per minute (rpm), to ensure the dissociation of the microorganisms from the surfaces. Then, the contaminated surfaces were washed by collecting and releasing the PBS 1× using a pipette at least four times. After this step, 10-fold serial dilutions in PBS 1× were performed and, finally, 1 mL of the undiluted sample and of each dilution was spread over the Petri dishes, and 15 mL of PCA was included. The colonies grown on PCA plates were counted according to the microbial strain incubation time at 35 ± 1 °C.Determination of antimicrobial activity (R) of enamel surfaces contaminated for 24 h: The antimicrobial efficacy of enameled surfaces with 1% wt. AgNO_3_ and 2% wt. AgNO_3_ was evaluated by assessing the recovered microbial cells in contact with the enameled surfaces included for 24 h. The general formula for calculating the recovered cells/cm^2^ from each specimen as per ISO 22,196 standard is as follows:

N (cells/cm^2^) = (100 * C * D * V)/A(1)
where N is the number of viable counts recovered per cm^2^, C is the colony count, D is the dilution factor of the counted plate, V is the volume of the PBS added to the test samples and A is the surface area (mm^2^) of the PP cover films. As per ISO standard accordance, three conditions should be satisfied, respectively, to have a valid test.
I.The average number of viable counts recovered at T0 from the control samples should range between 6.2 × 10^3^ cells/cm^2^ and 2.5 × 10^4^ cells/cm^2^.II.The logarithmic value of the number of viable counts recovered from control samples immediately after inoculation must meet the following criteria:

(L_max_ − L_min_)/L_mean_ ≤ 0.2(2)
where L_max_ is log10 of the maximum number of viable microbial counts, L_min_ is log10 of the minimum number of viable counts, and L_mean_ is the mean number of viable counts recovered from control samples.
III.The number of viable microbial counts recovered from each reference sample at T24 should be more than 6.2 × 10 cells/cm^2^.

When the conditions mentioned above are satisfied, the antimicrobial activity of test specimens is calculated by the following formula:R = (U_t_ − U_0_) − (A_t_ − U_0_) = U_t_ − A_t_(3)
where R represents antimicrobial activity, U_0_, and at log10 of the number of viable microbial counts recovered from control samples at T0, Ut represents an average of the log10 of the number of viable counts recovered at T24 from control samples, and A_t_ is the average of the log10 of the number of viable counts recovered from treated samples at T24.

The residual microbial counts at the different time points (T0, T2, T8, and T24 for *E. coli* and *S. aureus* and T24, T48, and T72 for *C. albicans*) were determined by calculating the CFU/mL.

## 5. Conclusions

The persistence of nosocomial pathogens on inanimate surfaces in hospital and healthcare settings is a major source of transmission of HAIs. The highest likelihood of lengthy persistence is provided by a large inoculum, a high relative humidity (such as more than 70%), and low temperatures [67], which are addressed through this study with the ISO:22196 standard.

The functionalized enameled surfaces were supplemented with 1% wt. AgNO_3_ and 2% AgNO_3_ have significant antimicrobial activity. The experiments carried out in the present study confirmed that the enameled coated surfaces containing Ag have a complete antibacterial effect after 24 h of exposure against the Gram-positive *S. aureus* and the Gram-negative *E. coli*. Moreover, the treated enameled surfaces E2, containing more Ag, were more effective in reducing the viability of both bacteria than the treated enameled surfaces E1 at earlier time intervals. With respect to *E. coli*, *S. aureus* showed less reduction in the number of bacteria during the time course, likely due to its thicker cell wall. Nevertheless, longer exposure to silver-based coated surfaces led to complete inhibition of *S. aureus*. Furthermore, this work proves an antifungal activity (fungistatic or fungicidal) of the silver-based enameled coatings against the opportunistic pathogenic fungus *C. albicans*.

In the future, these silver-based enameled coatings could be tested also against other nosocomial clinical isolates that are multi-drug resistant (MDR). Due to their unique properties, such as non-toxicity, hygiene, ease of cleaning, resistance to corrosion and abrasion, surface smoothness, hardness, and durability, silver-based enameled coatings can be applied in the field of architecture and design, and in the medical, chemical and pharmaceutical areas that can benefit from the above-mentioned properties.

## Figures and Tables

**Figure 1 ijms-24-02364-f001:**
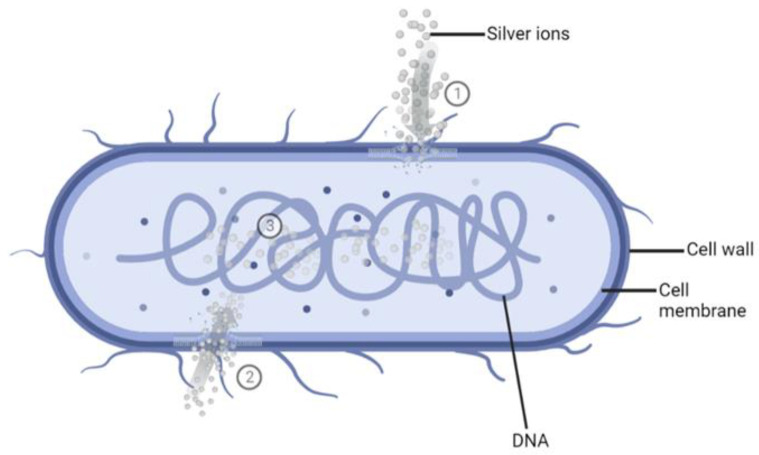
Schematic representation of the mode of action of silver ions (Ag^+^). ①. The binding of Ag^+^ to bacterial cell walls and membranes blocks the transport of essential substances. ②. Cellular uptake of Ag^+^ leads to the binding and inactivation of enzymes. ③. The interaction of Ag^+^ with bacterial DNA and blocking its cell division. Created with BioRender.com.

**Figure 2 ijms-24-02364-f002:**
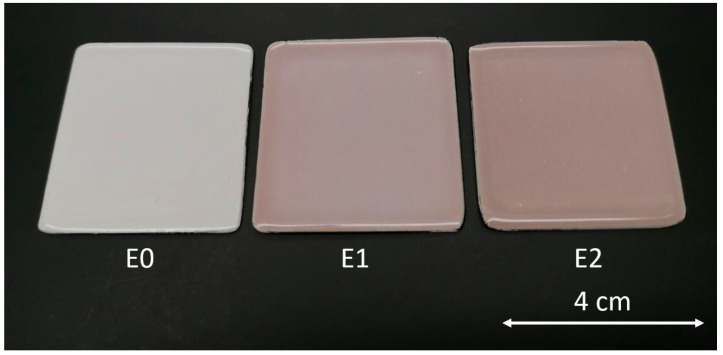
Top view of the enameled samples (40 mm × 40 mm). E0 (reference samples), E1 (admixed with 1% wt. AgNO_3_), and E2 (admixed with 2% wt. AgNO_3_).

**Figure 3 ijms-24-02364-f003:**
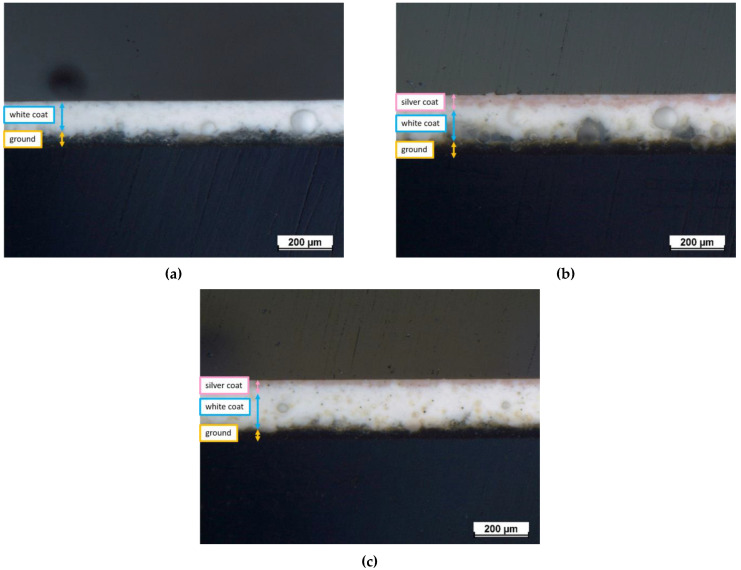
Optical cross-sectional micrographs of (**a**) reference sample E0, (**b**) sample E1, and (**c**) sample E2.

**Figure 4 ijms-24-02364-f004:**
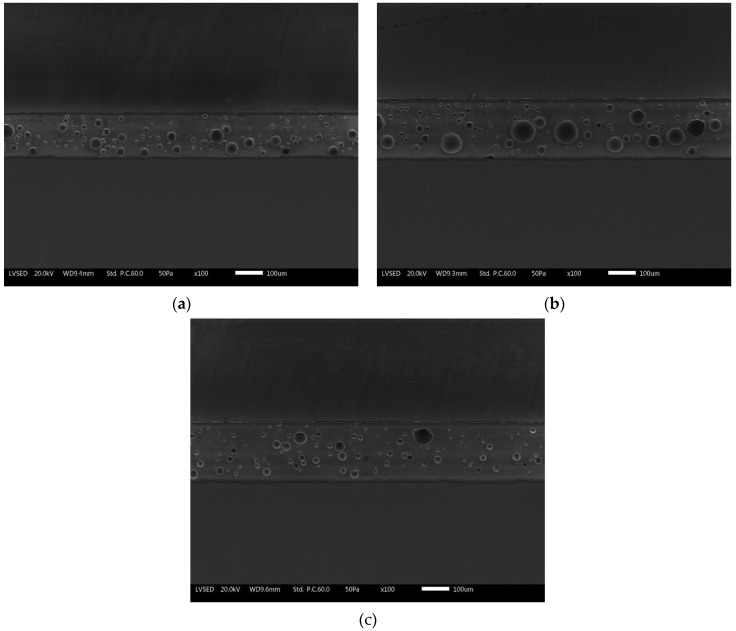
SEM cross-sectional micrographs of (**a**) reference sample E0, (**b**) sample E1, and (**c**) sample E2.

**Figure 5 ijms-24-02364-f005:**
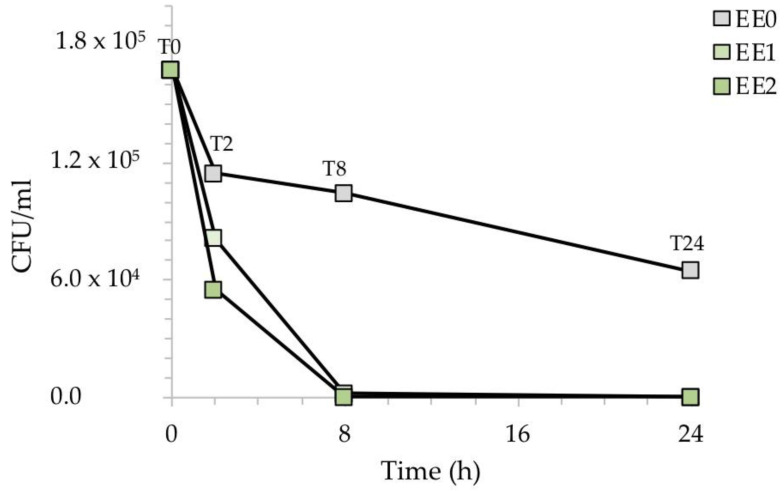
CFU/mL recovered after the exposure of the silver-treated contaminated surfaces to *E. coli* with respect to the untreated control. The results are representative of three independent determinations.

**Figure 6 ijms-24-02364-f006:**
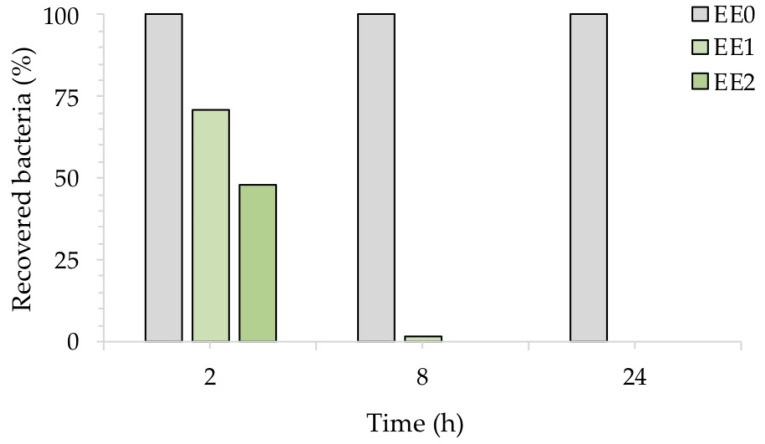
Antibacterial activity of the test surfaces against *E. coli* after, 2, 8 and 24 h of exposure.

**Figure 7 ijms-24-02364-f007:**
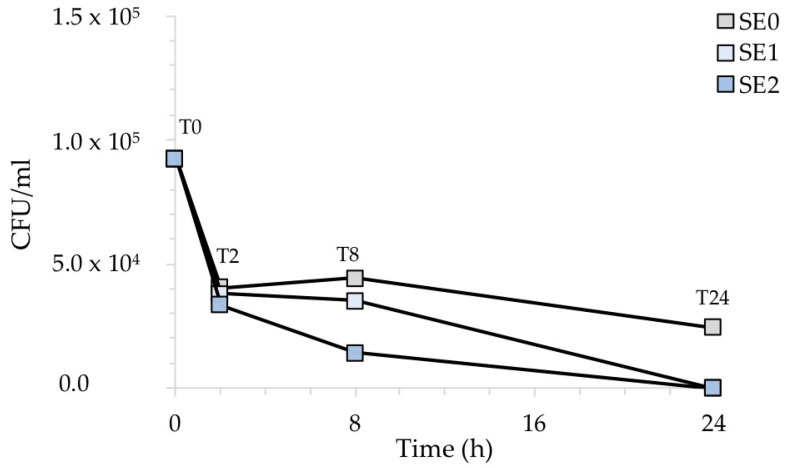
CFU/mL recovered after the exposure of the silver-treated contaminated surfaces to *S. aureus* with respect to the untreated control. The results are representative of three independent determinations.

**Figure 8 ijms-24-02364-f008:**
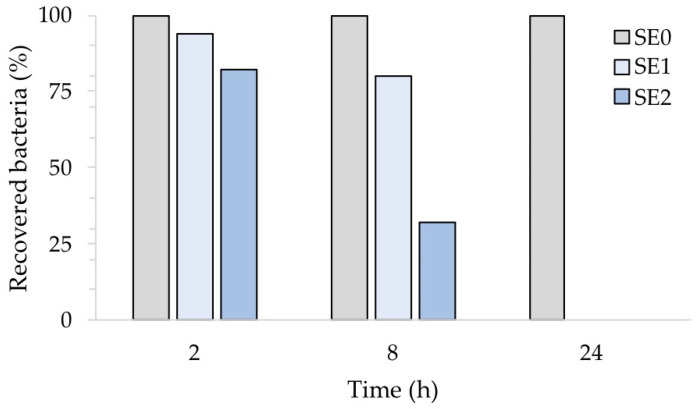
Antibacterial activity of the test surfaces against *S. aureus* after 2, 8 and 24 h of exposure.

**Figure 9 ijms-24-02364-f009:**
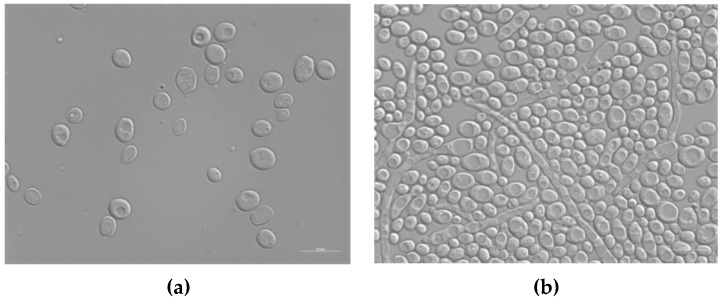
Differential interference contrast (DIC) micrographs showing *C. albicans* at T0 (**a**) and T24 (**b**) of contact with the surface of the untreated enameled coating E0. Size bar, 10 μm.

**Figure 10 ijms-24-02364-f010:**
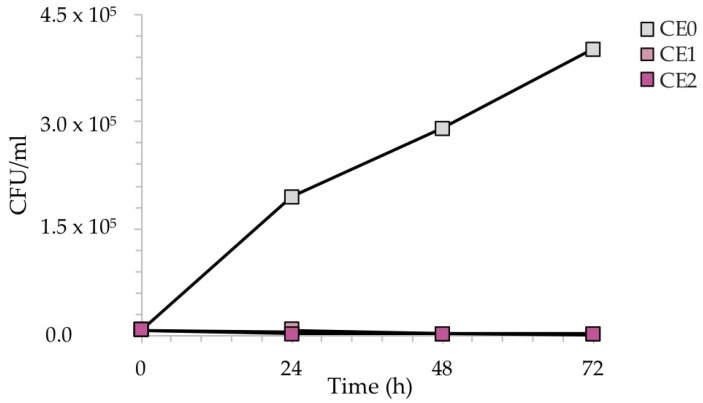
CFU/mL recovered after the exposure of the contaminated silver-treated enameled surfaces to *C. albicans*. The results are representative of three independent determinations.

**Figure 11 ijms-24-02364-f011:**
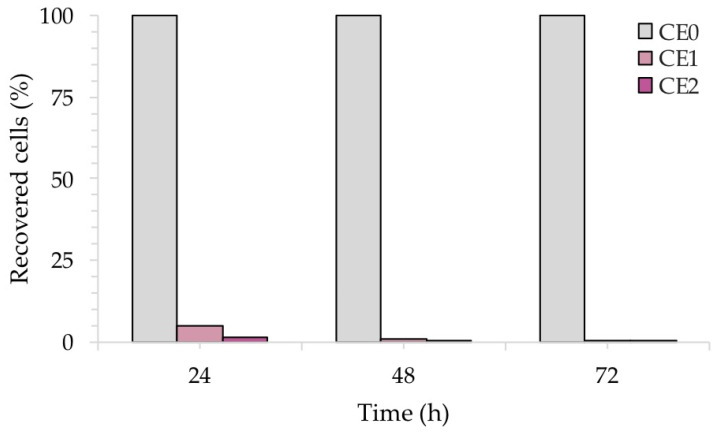
Antifungal activity of the test surfaces against *C. albicans* after 24, 48 and 72 h of exposure.

**Figure 12 ijms-24-02364-f012:**
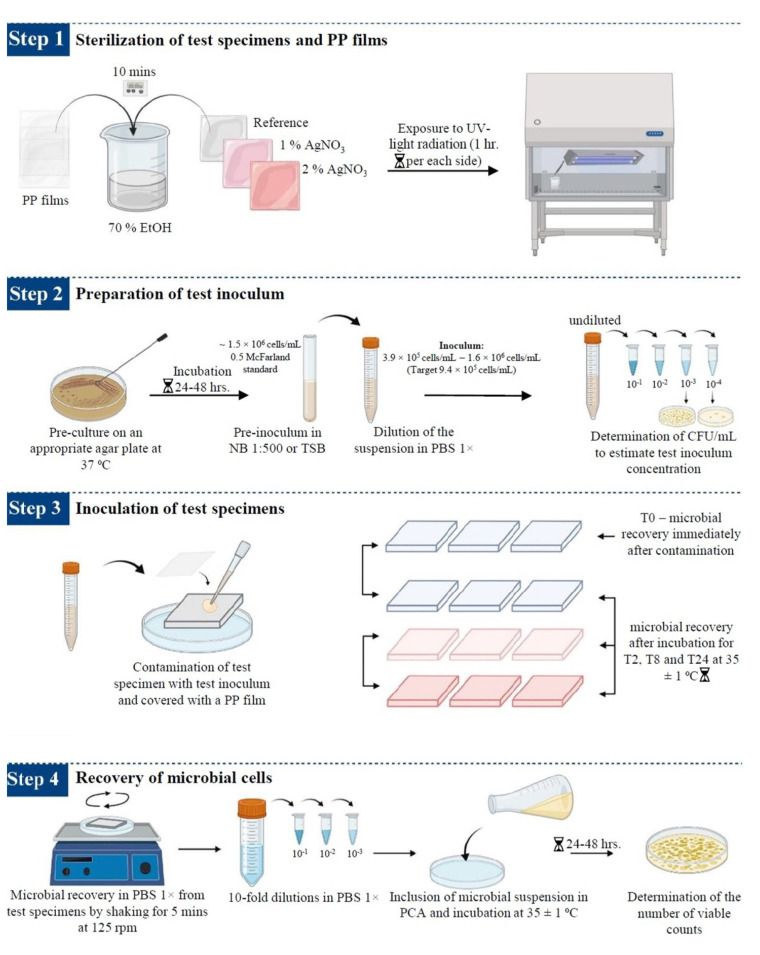
Schematic representation of the procedure used to determine the antimicrobial activity of the enameled surfaces. The image was taken and modified from [20].

**Table 1 ijms-24-02364-t001:** Surface roughness measurements (in-plane measurements).

Sample Type	Ra (μm)	Rz (μm)
E0	0.15 ± 0.01	0.74 ± 0.05
E1	0.15 ± 0.02	0.77 ± 0.16
E2	0.11 ± 0.02	0.63 ± 0.09

**Table 2 ijms-24-02364-t002:** Sample thickness measurements.

Sample Type	Total Thickness (µm)	Ground Coat (µm)	White Coat (µm)	Silver Coat (µm)
E0	171 ± 1	51 ± 7	120 ± 6	N/A
E1	217 ± 3	50 ± 5	137 ± 1	30 ± 2
E2	214 ± 1	39 ± 7	151 ± 3	24 ± 3

**Table 3 ijms-24-02364-t003:** Samples nomenclature.

	Reference Sample	1 wt.% AgNO_3_	2 wt.% AgNO_3_
**Sample nomenclature**	E0	E1	E2
**Number of layers**	2	3	3

## Data Availability

The data presented in this study are available on request from the corresponding author. The data are not publicly available due to the absence of an institutional repository.

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
