# Peer review of "Antimicrobial Performance of Innovative Functionalized Surfaces Based on Enamel Coatings: The Effect of Silver-Based Additives on the Antibacterial and Antifungal Activity"

_ijms, 2023, doi:10.3390/ijms24032364_

Round 1

Reviewer 1 Report

1. Fig. 1 can be moved to SI.

2. What is the recyclability of the coating.

3. What is the Ag status in the coating for antibacterial test, Ag+ or Ag0?

4. What is the Ag status in the coating after antibacterial test?

5. What is the surface morphology of the coating?

6. What about the fastness of the coating?

Author Response

  1. Fig. 1 can be moved to SI.

Authors: We would prefer to keep the figure in the text, as it explains the action of the silver ions in a simple way, perhaps not clear to all readers, and we don't show other supplemental material.

  1. What is the recyclability of the coating.

Authors: We thank the reviewer for raising this good point. Vitreous enamel coatings, despite being based on a relatively ancient production technique, meet very modern criteria of environmental friendliness and recyclability. At the end of life, enamelled metallic components can be treated as metal scraps, without the need to separate the coating from the metal substrate, as enamel compounds become smelt waste that separates from the metal and does not alter the properties of the smelt itself [1]. In addition to that, many recent technological advances have allowed to improve the closed-loop waste management in the enamelling process. In this context, the reduction of harmful chemicals and the introduction of new deposition methods have allowed the full recycling of effluents and the recycling of wet and dry formulation with a consequent reduction of oversprayings [2,3]. Thus, the production of enamel coatings is perfectly compatible with the efforts to promote circular economy and the closed-loop management plan.

This information, together with the new references, has been included in the introduction of the revised manuscript (lines 85-96).

[1] Shippy, G., Reclamation of Scrap Frit. In W.J. Smothers (Ed.), Proceedings of the 41st Porcelain Enamel Technical Forum: Ceramic Engineering and Science Proceedings (1979).

[2] Elliott, R.H., Jr., Zero Discharge, Zero Pollution, and Source Reduction. In J.B. Wachtman (Ed.), Proceedings of the 50th Porcelain Enamel Institute Technical Forum: Ceramic Engineering and Science Proceedings (1989).

[3] Cameron, D.S., Waste Minimization. In J.B. Wachtman (Ed.) Proceedings of the 50th Porcelain Enamel Institute Technical Forum: Ceramic Engineering and Science Proceedings (1989).

  1. What is the Ag status in the coating for antibacterial test, Ag+ or Ag0?

Authors: Unfortunately, the authors have not had the opportunity to run an XPS measurement, as they are not equipped with the appropriate instrumentation. However, partial reduction to Ag0 is likely to occur at the high temperatures of the thermal process (800°C), to which the samples are subjected. This assumption is confirmed by the color of the samples: the particles of Ag0 lead to a reduction of the reflected light with consequent darkening of the coating. Thus, samples E2 appear darker than samples E1, and both clearly darker than samples E0 without silver (Fig 3).

  1. What is the Ag status in the coating after antibacterial test?

Authors: Again, the authors have not had the opportunity to perform an XPS measurement. However, no changes of the Ag status are expected after the antibacterial test. Indeed, neither the culture medium (highly diluted) nor the neutral PBS solution of the antibacterial test are expected to alter the nature of the silver. Consistently, no color change of the samples is observed following the antibacterial tests. Thus, as mentioned in the previous comment, it is highly probable that some of the silver in the coating is in the form of Ag0.

  1. What is the surface morphology of the coating?

Authors: The authors are not completely clear of what is intended by this question. We have already extensively discussed the surface morphology of the samples, presenting surface roughness analyses and showing the sections of the samples, highlighting the various layers that compose them. Moreover, many 'superficial' aspects of the samples have already been treated in previous work [20].

  1. What about the fastness of the coating?

Authors: Also these aspects have been already treated in the previous work published by the authors [20]. The coatings show good resistance towards mild abrasive processes also when coupled with aggressive chemical solutions. In addition, the presence of a considerable amount of residual silver on the surface makes this type of coatings still able to provide a good antibacterial activity after mechanical damage. Instead, purely 'mechanical' performance has never been investigated, as the authors focused on the durability of the coating and its antibacterial properties after accelerated degradation.

Author Response

  1. How can this film be applied to a given surface?

Authors: Paragraph 2.1 describes the production of the samples with the deposition of the layers. The deposition takes place by wet spraying method, therefore it can be applied to substrates of different geometries and dimensions. Consequently, the process can also be scaled to an industrial level without plant problems.

  1. Can the colony-forming units per ml (CFU/ml) be reduced or completely extinguished? In fact, Figs. 6 − 10 are extremely interesting since they show a reduction to zero for CFU/ml.

Authors: Yes, the CFU/ml are progressively reduced and completely extinguished over time, consistently with no detection of CFU/ml at 24 h.

  1. Fig. 3 displays the surfaces of three samples without, with 1 %, and with 2 % Ag+ incorporated. The treatment was a combustion at 800°C. At these temperatures, the Ag+ is expected to be reduced to elemental Ag0 to a small extent.1 Nanoparticles of Ag0 are generated which can either scatter or absorb light. In both cases, this leads to a reduction of the reflected light which causes this darkening.

Authors: The authors fully agree with the reviewer. In fact, the high temperatures of the process and the color change of the 3 different samples suggest the partial presence of Ag0.

  1. Figs. 6 + 8 clearly show the drop of CFU/ml. But the graphs can be made more beautiful. Please, consider a grid like this displayed in Fig. 1. No more than 5 numbers at each axis, big numbers (I use pt 28).

Authors: We have revised the graphs (Figs.6 + 8 and also 11-12 for consistency) with no more the 5 numbers for each axis and larger font for the numbers and the axis names. The authors agree that now they look more appealing and thank the reviewer for his/her suggestion.

  1. Roughness is measured using an AFM. The paper would gain in quality and importance if the authors could one or two micrographs. AFM are now in wide-spread use, and it should easily be manageable.

Authors: Certainly micrographs acquired in AFM would be representative of an accurate 'scientific' work. However, they probably would not add significantly more additional information. The authors deal with a coating, a material that could be defined as 'macroscopic'. For this reason we preferrred to acquire macroscopic level information (such as the roughness obtained with a roughness gauge) rather than microscopic data relating to small areas (such as the results provided by AFM analyses). Furthermore, unfortunately, the authors do not have the equipment available and so they would not be able to provide this data.
